# Recent Imaging Updates and Advances in Gynecologic Malignancies

**DOI:** 10.3390/cancers14225528

**Published:** 2022-11-10

**Authors:** Taher Daoud, Sahil Sardana, Nir Stanietzky, Albert R. Klekers, Priya Bhosale, Ajaykumar C. Morani

**Affiliations:** Department of Diagnostic Radiology, The University of Texas MD Anderson Cancer Center, 1515 Holcombe Blvd., Houston, TX 77030, USA

**Keywords:** gynecologic malignancies, functional imaging, imaging in gynecologic malignancies

## Abstract

**Simple Summary:**

Gynecological malignancies are among the most common cancers with significant morbidity and mortality worldwide. Management and overall patient survival is reliant upon early detection, accurate staging and early detection of any recurrence. This article provides a comprehensive review of the recent advances in imaging of gynecologic malignancies with emphasis on cervical, endometrial, and ovarian neoplasms.

**Abstract:**

Gynecologic malignancies are among the most common cancers in women worldwide and account for significant morbidity and mortality. Management and consequently overall patient survival is reliant upon early detection, accurate staging and early detection of any recurrence. Ultrasound, Computed Tomography (CT), Magnetic resonance imaging (MRI) and Positron Emission Tomography-Computed Tomography (PET-CT) play an essential role in the detection, characterization, staging and restaging of the most common gynecologic malignancies, namely the cervical, endometrial and ovarian malignancies. Recent advances in imaging including functional MRI, hybrid imaging with Positron Emission Tomography (PET/MRI) contribute even more to lesion specification and overall role of imaging in gynecologic malignancies. Radiomics is a neoteric approach which aspires to enhance decision support by extracting quantitative information from radiological imaging.

## 1. Introduction

Gynecological malignancies are among the most common cancers worldwide [1]. Cervical cancer is the 3rd most common female malignancy and is the leading cause of cancer related death in young women [1]. Endometrial carcinoma is the most common gynecologic malignancy, with nearly 40,000 newly diagnosed cases yearly in the United States [2]. Ovarian cancer is less common yet lethal disease with 19,880 newly diagnosed women in the United States and nearly 12,810 deaths related to this cancer in 2022 [3].

The recent FIGO (Fédération Internationale de Gynécologie et d’Obstétrique- The International Federation of Obstetrics and Gynecologic) staging updates acknowledges the role of imaging including the molecular imaging for accurate staging of gynecological malignancies. For example, revised FIGO staging for cervical cancer now enables the upstaging of the cancer cervix to stage IIIC based on radiological (not only pathological) detection of pelvic +/− paraaortic lymphadenopathy [4].

In this review article we will discuss the recent advances in imaging of gynecologic malignancies with emphasis on cervical, endometrial, and ovarian neoplasms.

## 2. Methods

Literature search done about recent imaging updates and advances in gynecologic malignancies (with emphasis on endometrial, cervical and ovarian neoplasms) including ultrasound, cross-sectional imaging, hybrid imaging and Artificial intelligence, deep learning, Radiomics and Radiogenomics. Representative images examples were collected from our institutional data base. As it is a retrospective review article with HIPPA compliant anonymous images, patient consent is not applicable and waived per MD Anderson Institutional Review Board.

## 3. Discussion

Ultrasound is usually the primary modality to evaluate women with pelvic symptoms. It has the advantages of wide availability, feasibility, and cost-effectiveness with no radiation hazards. Limitations include operator dependence, limited field of view, and low contrast resolutionMRI is regarded as the gold standard for local delineation of most gynecological malignancies owing to superb soft tissue contrast and resolution without exposing the patients to ionizing radiations. In addition, DCE and DWI provide additional data regarding tissue perfusion and cellular density, respectively.FDG PET-CT is a modality of high specificity yet low sensitivity in delineating primary gynecologic malignancies with a well-established role in preoperative staging, monitoring therapy response, and detecting recurrence in all locally advanced gynecological malignancies. New tracers are introduced but with limited data in the literature, thus not routinely used in clinical practice.PET-MRI allows a one-stop assessment of gynecological malignancies, combining the functional and quantitative metabolic data of PET with the high-resolution anatomic and functional imaging properties of MRI. Although it shows promising diagnostic performance, limited data in the literature exists regarding the justification of its high cost compared to separately acquired PET-CT and MRI.Radiomics and radiogenomics are promising approaches that may aid the diagnosis, prognosis, and assessment of treatment response in gynecological malignancies. However, there is a need for more extensive prospective studies and standardization of relevant imaging biomarkers before they can be applicable to the clinical workflow.

## 4. Imaging

### 4.1. Ultrasound

Ultrasound is usually the modality of choice to evaluate women with pelvic symptoms. It may be useful in detecting primary gynecologic malignancy (ovarian, endometrial or cervical), and has an added advantage of wide availability, feasibility, cost effectiveness and is without radiation hazards. The routine ultrasound for the evaluation of the uterus, ovaries and adnexa can be performed using transvaginal, transabdominal, or transperineal approach or combination of techniques. Continuous advances in this field are being introduced in an attempt to overcome its limitations (being operator dependent, limited field of view and rather low contrast resolution) and to enhance its diagnostic value [5]. Doppler ultrasound is used for the assessment of vascularity in various pelvic pathologies.

Regarding myometrial neoplasms, ultrasound is widely used for the detection and follow up of leiomyomas (which are the most common uterine tumors). They classically appear as well-defined hypoechoic masses (Figure 1) whereas internal areas of variable echogenicity might be seen in degenerated leiomyomas (cystic or hemorrhagic degeneration). There are no definitive sonographic feature that help differentiate leiomyomas from uterine sarcomas. However, marked central vascularity on spectral ultrasound Doppler makes the diagnosis of uterine sarcoma more likely than degenerated leiomyoma. Some sonographic features that may be helpful in differentiating uterine sarcomas from leiyomyomas include large size (>10 cm), irregular margins, marked heterogeneity, extensive intrinsic hemorrhage and associated ascites [6,7].

Ultrasound is commonly used to assess abnormal uterine bleeding. In these patients ultrasound may reveal diffuse or focal endometrial thickening (e.g., hyperplasia, carcinoma), and detect endometrial masses such as polyps and carcinoma.

The classic sonographic appearance of a polyp is a well-defined homogenous hyperechoic mass which may show a single feeding vessel on Doppler (pedicle artery sign which is sensitive and specific sign). Saline-infused Sonohysterography can be used to assess polyps in the setting of diffuse endometrial thickening and has a better sensitivity and specificity [8].

Endometrial cancer: The ultrasound features suggestive of endometrial carcinoma are thickening of the endometrial lining with indistinct/irregular interface between the endometrium and myometrium, heterogeneity of the endometrium and increased endometrial vascularity on Doppler [7,9]. Sometimes there may be an overlap of imaging findings of endometrial polyps, hyperplasia and endometrial carcinoma. In the setting of abnormal bleeding, biopsy is usually performed even with mildly increased endometrial thickness, i.e., >16 mm in premenopausal and 5 mm in post-menopausal females [7,10]. Sonohysterography is reported to have better sensitivity than transvaginal ultrasound (TVUS) in detection of myometrial invasion in cases of endometrial cancer, though it also carries a post procedural risk of malignant cell dissemination [6,11]. Studies reported a 71–85% sensitivity, 72–90% specificity and 72% to 84% accuracy for ultrasound in detecting myometrial invasion by endometrial carcinoma while in the detection of cervical invasion by endometrial carcinoma, sensitivity was 29% to 93%, specificity 92% to 94%, and accuracy 78% to 92% [7,12,13,14].

To improve the ultrasound diagnostic accuracy, Contrast enhanced ultrasonography (CE-US) can be used, in which a microbubble contrast agent is intravenously injected aiming to improve the overall imaging quality via the direct detection of tumor neovascularity (as it has the capability of detection of vessels with diameters less than 0.1 mm which are beyond the scale of color Doppler and power Doppler ultrasound), thus combining visualization of both the tumor morphology and vascularity.

CE-US in combination with 2D TVUS improves the detection of deep myometrial invasion as well as cervical stromal involvement in cases of endometrial cancer, however it can be limited by patients’ body habitus. Moreover, it can improve the accurate disease classification as more advanced cases of endometrial cancer are associated with certain qualitative parameters (endometrial focal filling & focal wash out patterns and prior wash in pattern of contrast compared to myometrium) as well as semi quantitative parameters (wash-in slope, shorter time-to-peak, higher peak intensity and greater area under the time–intensity curve), and CEUS overall helps to reduce the unnecessarily extensive surgeries by about 10% [11].

Cervical cancer: Ultrasound examination is not recommended either for screening or staging of cervical carcinoma as early cervical carcinomas are usually small in size and isoechoic to the normal cervical mucosa and hence difficult to detect sonographically. Ultrasound can detect more advanced (invasive) cervical cancers with variable sonographic appearances that include area of altered echogenicity, haziness of the normal cervical zonal anatomy, distorted cervical outlines and iso/hypoechoic mass replacing the cervical tissue. More recent studies showed that advanced ultrasound techniques such as 3D ultrasound with color Doppler might improve the clinical application of ultrasound in cervical cancer for primary tumor detection, evaluation of local extension and to assess the response to treatment (especially when performed by trained personnel). Additionally, ultrasound might be helpful in prediction of the cervical cancer response to radiotherapy, chemotherapy or neoadjuvant chemotherapy (via monitoring tumor vascularity by Doppler as most of cervical carcinomas are hypervascular). Ultrasound has a limited role in determination of metastatic lymphadenopathy [7,15].

In ovarian tumors: Ultrasound features suspicious for malignancy in ovarian masses, include obvious solid components (mural irregularities, solid nodule, and papillary projection), increased vascularity on Doppler, internal septations with thickness ≥3 mm and presence of ascites, peritoneal or omental deposits [16].

CE-US is increasingly used for the detection and preoperative assessment of ovarian cancers as it increases the sensitivity of detection of nodal involvement compared to conventional 2D ultrasound. CE-US can be used in patients with contraindications for enhanced CT or MRI techniques such as iodinated contrast related hypersensitivity and renal insufficiency [17].

Apart from its operator dependence, one of the major concerns in the use of ultrasound is the lack of standardized terminology that might lead to significant incoherence in reporting among different reporters. As ultrasound is the most used modality in the primary evaluation of adnexal and ovarian pathologies, the Ovarian-Adnexal Reporting and Data System (O-RADS US) (Table 1) was recently established on ultrasound [18] to improve the reporting quality by providing consistent interpretation, attempt to eliminate the ambiguity in the ultrasound reports and enhance the communication between the ultrasound reporter and the treating physician. This may lead to better diagnostic accuracy and therefore, may positively influences the management and follow up of these patients [18].

**Table 1 cancers-14-05528-t001:** Ovarian-Adnexal Reporting and Data System (O-RADS) US risk stratification and management system.

O-RADS Score	Risk Category	Lexicon Descriptors			Management		
					Premenopausal		Post-Menopausal
0	Incomplete Evaluation	N/A			Repeat/Alternative Study		
1	Normal Ovary	Follicle is a simple cyst ≤ 3 cm			None	N/A	
		Corpus Luteum ≤ 3 cm					
2	Almost certainly benign (risk of malignancy < 1%)	Simple cyst	≤3 cm		N/A	None	
			3 m to 5 cm		None	Follow up in 1 year	
			>5 cm to <10 cm		Follow up in 8–12 weeks	Follow up in 1 year (at minimum)	
		Classic benign lesions	Typical hemorrhagic cyst		If >5 cm to <10 cm, follow up in 8–12 weeks	US specialist, gynecologist management or MRI	
			Typical dermoid cyst <10 cm (Figure 2)		Follow up in 8–12 weeks	US specialist, gynecologist management or MRI	
			Typical endometrioma <10 cm		Follow up in 8–12 weeks	US specialist, gynecologist management or MRI	
			Simple paraovarian cyst of any size		None	Single follow up in 1 year	
			Typical peritoneal inclusion cyst of any size		Gynecologist management		
			Typical hydrosalpinx of any size		Gynecologist management		
		Non-simple unilocular cyst, smooth inner margin	≤3 cm		None	Follow up in 1 year	
			>3 cm to <10 cm		Follow up in 8–12 weeks	US specialist or MRI	
3	Low risk of malignancy (1% to less than 10%)	Unilocular cyst ≥ 10 cm (simple or non-simple)			US specialist or MRI Gynecologist management		
		≥10 cm Typical dermoid/hemorrhagic cysts or endometrioma					
		Unilocular cyst with irregular inner wall <3 mm height (regardless cyst size)					
		Mutlilocular cyst < 10 cm with smooth inner wall, CS = 1–3					
		Solid smooth (regardless its size), CS = 1					
4	Intermediate risk of malignancy (10% to less than 50%)	Multilocular cyst with no solid component		≥10 cm, smooth inner wall, CS = 1–3	US specialist or MRI Management by GYN-oncologist or gynecologist with GYN-oncologist consultation		
				Any size, smooth inner wall, CS = 4			
				Any size, irregular inner wall and/or irregular septations, any color score			
		Unilocular cyst with solid component		Any size, 0–3 papillary projections, CS = any			
		Multilocular cyst with solid component		Any size, CS = 1–2			
		Solid		Smooth, any size, CS = 2–3			
5	High risk of malignancy (≥50%)	Unilocular cyst, ≥4 papillary projections, (regardless its size and CS)			GYN-oncologist management		
		Multilocular cyst with solid component (regardless its size), CS = 3–4					
		Solid smooth, CS = 4 (regardless its size)					
		Solid irregular (regardless its size and CS)					
		Ascites and/or peritoneal nodules					

CS = color score, GYN = gynecologic, N/A = not applicable.

**Figure 2 cancers-14-05528-f002:**
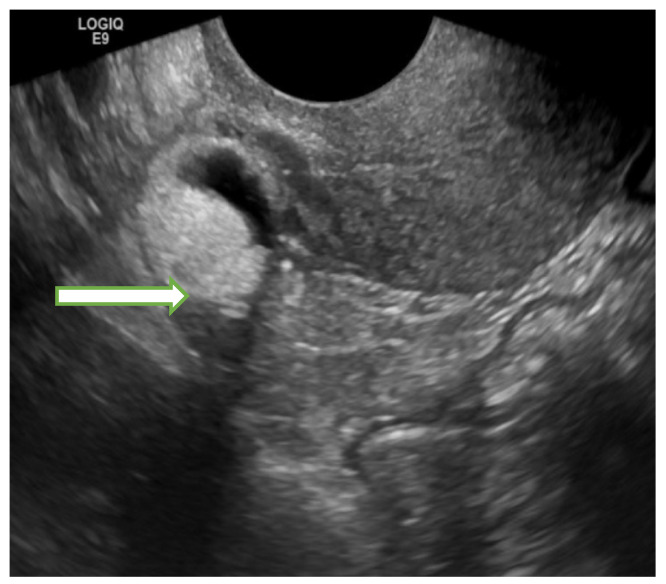
Transvaginal ultrasound image of the pelvis in a 55 years old lady shows a right adnexal 1.8 × 1.8 × 1.9 cm mass with mixed echogenicities (fat, calcification, and soft tissue) consistent with a dermoid cyst. The lesion remained stable in subsequent follow up studies confirming its benign nature.

### 4.2. Magnetic Resonance Imaging (MRI)

Contrast enhanced MRI of the pelvis is considered the gold standard for local staging of various gynecological malignancies as it provides superb soft tissue contrast and resolution allowing to precisely determine local tumor extensions without exposure of the patients to ionizing radiations.

While conventional MRI images can provide detailed information about the precise location, margin and local extension of the gynecologic tumors, more data regarding the tissue perfusion could be obtained using Dynamic contrast enhanced (DCE) studies and, the tissue cellular density along with mobility or diffusion of water protons can be assessed using Diffusion weighted imaging (DWI) on MRI [19].

Endometrial cancer: MRI can accurately assess depth of myometrial invasion, cervical involvement, and presence of metastatic lymph nodes that impacts the tumor grading and hence overall survival. Classically, endometrial cancer appears as localized or diffuse area of heterogeneously intermediate T2 signal (relative to the hyperintense T2 signal of the normal endometrium). Endometrial cancer usually shows restricted diffusion evidenced by high signal intensity on DWI (at high b value) and low ADC value in contrast with benign endometrial conditions such as endometrial polyps and hyperplasia that typically have significantly higher ADC value. On DCE imaging, small tumors may show early enhancement compared to the normal endometrium while the tumor classically appears hypointense relative to the myometrium during the later phases of enhancement [20]. Myometrial invasion or cervical stromal invasion is better evaluated in in the equilibrium phase and delayed phase, respectively on the DCE images. The presence of intact low T2 signal of the junctional zone mostly excludes the myometrial invasion. Less than 50% myometrial thickness invasion represents stage 1A tumor, whereas more than 50% myometrial thickness invasion indicates 1B tumor, and thus affecting the management. T2 WI can be used to differentiate endometrial cancer from submucosal leiomyoma which classically elicits low T2W signal and shows intense early phase contrast enhancement (Figure 3, Figure 4 and Figure 5).

Cervical cancer: MRI is crucial in the evaluation of cervical cancer allowing for accurate determination of the size, location, extent of the tumor including any parametrial, pelvic side wall or nodal involvement. MRI may be very helpful in selecting patients who might be eligible for conservative fertility preservation procedures, particularly for young women with FIGO 1B1 and 1B2 tumors. Typically, cervical tumors elicit intermediate to high signal on T2 WI compared to myometrium, and show restricted diffusion. On DCE, the tumor usually shows early avid enhancement compared to unaffected cervical tissue. Enhancement is frequently heterogeneous in large tumors secondary to necrosis, while smaller tumors elicit more uniform enhancement. A rather uncommon subtype is the Adenoma malignum which typically appears as multicystic lesion with enhancing solid component. Careful evaluation of the integrity of the cervical stromal rim which is usually >3 mm in thickness and normally appears as homogenous hypointense T2 WI rim, is needed to exclude parametrial involvement. Parametrial involvement might appear as soft tissue speculations or nodularities in the parametrium, or as diffusely thickened (>3 mm) cervical rim with inhomogeneous T2 signal [21]. MRI can reliably differentiate between local recurrence and post radiotherapy changes. Local recurrence typically appears as mass of intermediate to high signal on T2WI, showing early contrast enhancement on DCE and diffusion restriction, while post radiation changes/fibrosis typically shows no diffusion restriction and show either no or late enhancement on DCE [22,23] (Figure 6 and Figure 7).

Ovarian tumors: MRI is commonly used to assess indeterminate adnexal masses seen on ultrasound. MRI features suggestive of malignancy include large solid component, early mass enhancement on DCE, mural or septal thickness of >3 mm, internal mural nodularities, presence of necrosis within the mass, extension to other pelvic organs, mesentery, omentum, lymph nodes and presence of ascites [24]. MRI with the superior contrast and soft tissue resolution can differentiate benign from malignant adnexal masses. A dermoid will have a high signal on the T1 and T2 FSE weighted images and will show decreased signal in the fat suppressed sequences [24,25] (Figure 8 and Figure 9). A relatively novel application of MRI in ovarian tumors is the Ovarian-Adnexal Reporting and Data System (O-RADS) MRI risk score that implies a codified scoring system to assess the malignant potential of ovarian and adnexal lesions based on MRI imaging characters (lesion composition, signal characters and enhancement pattern) [26] (Table 2).

Lesion composition is either cystic or solid. Cysts can be unilocular/multilocular with/without solid components and simple/nonsimple cyst (non simple cyst might have endometriotic, hemorrhagic, proteinaceous, or lipid contents). Lesion is considered solid if composed of at least 80% enhancing solid tissue [26].

The signal intensity is described as homogeneous/heterogeneous and hypointense/intermediate/hyperintense on T2-weighted images (in relation to iliopsoas muscle and urine or cerebrospinal fluid) and T1-weighted images (in relation to the iliopsoas muscle and fat). At high b-value diffusion-weighted imaging (DWI), the signal intensity is described as low or high (in relation to urine or cerebrospinal fluid).

### 4.3. Computed Tomography (CT)

CT scan has several advantages over other modalities. Compared to MRI, it is relatively more available, has shorter image acquisition time and the possibility of rapidly obtaining multiplanar reconstructed images. Compared to ultrasound, CT has wider field of view, higher spatial resolution and is less operator dependent. For evaluation of common gynecologic cancers by CT, intravenous contrast administration is usually required and rectal contrast can also be considered on a case by case basis. However, owing to its poor soft tissue contrast, CT scan has a subordinate role in local staging of the gynecologic malignancies. Instead, CT scan can be readily useful in detection of tumoral extension to any adjacent organ, pelvic and extrapelvic lymphadenopathy as well as distant metastases, e.g., peritoneal and pulmonary metastasis [27].

Disadvantages of CT include exposure of patients to ionizing radiation, possible adverse reactions to the contrast agents and iodinated contrast-limitations or contraindications in patients with renal insufficiency [27].

An emerging technology is Dual-energy CT (DECT) in which the datasets are simultaneously acquired from two different photon energies in a single CT acquisition. The differences in material composition can be detected on the basis of differences in photon absorption, which might be helpful in the evaluation of primary tumors and metastatic disease in patients with gynecologic malignancies. DECT may be of value in the assessment of some features suggestive of malignancy in cystic ovarian masses, for example internal septations more than 3 mm in thickness, intramural nodularities and papillary projections, etc. There are other potential applications for DECT in gynecological malignancies such as ovarian cancers, endometrial cancers and uterine sarcomas. DECT may increase the accuracy of staging of ovarian cancer by improving the identification of subdiaphragmatic tumoral implants prior to debulking surgery as well as easier detection of perihepatic and perisplenic implants. In uterine sarcomas it may help in the better assessment of musculoskeletal and hepatic metastatic lesions (as hepatic metastatic lesions may occasionally have attenuation similar to the liver rendering them undetectable on portal venous phase imaging). DECT could also be helpful in the detection of calcified metastatic lesions from papillary serous endometrial cancer [28,29,30].

### 4.4. Molecular Imaging

The role of molecular imaging in the early stages of gynecologic malignancies is limited compared to MRI and CT that are routinely used in local staging of the disease. PET has been shown to have higher accuracy in advanced disease evaluation, in comparison to MR and CT alone, and might help spare unnecessary surgical interventions in some patients [4].

The recent FIGO (Fédération Internationale de Gynécologie et d’Obstétrique- The International Federation of Obstetrics and Gynecologic) staging updates acknowledges the role of imaging including the molecular imaging for accurate disease staging. For example, revised FIGO staging for cervical cancer now enables the upstaging of the cancer cervix to stage IIIC based on radiological (not only pathological) detection of pelvic +/− paraaortic lymphadenopathy [4].

The National Comprehensive Cancer Network (NCCN) suggests considering FDG-PET/CT in all cases of gynecologic malignancies if metastasis is suspected at initial staging and for evaluation of suspected recurrence [4].

### 4.5. Positron Emission Tomography-Computed Tomography (PET-CT)

Hybrid imaging using [fluorine-18]-fluoro-2-deoxy-D-glucose positron emission tomography (18F-FDG PET CT) has established role in staging and monitoring of therapy as well as detection of recurrence in gynecologic malignancies with its ability to incorporate the anatomical data (CT) and the metabolic information (functional PET) (Figure 9 and Figure 10). Moreover, it has an important role in determining the metabolically active gross tumor volume (GTV) which includes the gross palpable or visible/demonstrable extent and location of malignant growth and thus helps in planning for radiation therapy. It also improves the detection of distant and nodal metastatic disease compared to CT alone [31].

However, high radiotracer uptake is not only seen in the malignancies but may also be seen with other benign lesions such as leiomyoma, pelvic inflammatory changes or with physiological conditions such as focal ureteric excretion mimicking active lymph node, physiological uptake by normal endometrium and normal ovaries in different menstrual phases. On the contrary, false negative results could be encountered in some scenarios, for example necrotic lymph nodes might show no or very low tracer uptake and physiological excretion of the tracer in the bladder might conceal small underlying masses. PET-CT is a modality of high specificity yet of low sensitivity in detection of primary lesion in gynecologic malignancies, and this may be challenging for the reporting radiologist who should be knowledgeable about its pitfalls to avoid faulty interpretations [32].

Endometrial cancer: FDG PET CT has well established role in detection of extrauterine spread before salvage radiotherapy or surgery. There are some promising studies [33,34] regarding the possible use of FDG PET CT as a predictive biomarker, however more improvements in the standardization and validation of metabolic threshold values are needed [32]. There is no established role for FDG PET CT in the detection, local staging or in the preoperative nodal staging of endometrial cancer [23,32].

Cervical cancer: FDG PET CT is the modality of choice for staging of advanced cervical cancer with better detection of distant and nodal metastasis. Moreover, in locally advanced cervical cancer, FDG PET CT may be used for staging, planning for radiotherapy and assessing the treatment response [4,32].

Ovarian cancer: FDG PET CT has established role for detecting suspected recurrence in patients with rising tumor markers (such as CA125) who could be candidates for salvage surgery or radiotherapy. There is no established role for FDG PET CT in assessing adnexal masses of indeterminate nature or for the primary staging of ovarian cancer. [23,24,32]. Although FDG PET CT is superior to CECT for detection of nodal and extraabdominal metastatic lesions, the studies have proven that this superiority is not reflected on the patients’ management [32,35,36].

Recently DOTATATE-PET/CT is proven to be of value in the work up of uncommon neuroendocrinal variants of ovarian and cervical malignancies [4].

A common feature of most of solid tumors is hypoxia. The hypoxic state induced by the tumor stimulates angiogenesis, disrupts the internal cellular environment and eventually may lead to increased incidence of tumor resistance, metastasis and recurrence. Thus hypoxia is a negative prognostic factor and a potentially useful prognostic biomarker in cancer patients [37].

Several image-based modalities for hypoxia evaluation exist. The most widely used is PET-based hypoxia imaging [38]. Advantages of PET-based hypoxia imaging include highest specificity to tissue hypoxia, good intrinsic resolution, tolerance by the patients, potential repetition and possibility of semiquantification/quantification of the hypoxic tumor burden [39]. A variety of hypoxia PET tracers have been developed and evaluated such as 2-nitroimidazole based (including FMISO, FAZA, FETNIM, FETA, EF1, EF3 and EF5) and Cu-ATSM [40].

### 4.6. PET-MRI

Hybrid imaging with Positron emission tomography-magnetic resonance imaging (PET-MRI) aims to combine the functional and quantitative metabolic data of PET with the high resolution anatomic and functional imaging properties of MRI in one combined or hybrid modality. It has the advantage of providing precise simultaneous evaluation of the primary gynecologic tumor and any possible metastasis with radiation exposure less than PET-CT by about 45% [41].

Despite the limited data in the literature, the diagnostic performance of PET-MRI for gynecological cancers appear promising compared to other imaging modalities. Several reports suggest that PET-MRI is superior to FDG PET CT in the assessment of local disease spread. PET-MRI is expected to play a promising role in planning for radiotherapy and assessing for treatment response in the future [19,42].

Endometrial cancer: SUV and ADC values are potential prognostic functional biomarkers. PET-MRI is comparable to PET-CT for primary tumor identification and detection of tumor recurrence (Figure 11).

Cervical cancer: PET-MRI potentially has a higher diagnostic sensitivity for local recurrence of cervical cancer than PET-CT. PET-MRI may also be of value in the detection of neuroendocrine cervical cancers. Moreover, the aggressiveness of cervical cancer could be correlated to the functional biomarkers like SUV and ADC values (Figure 10, Figure 12 and Figure 13). PET-MRI may also be valuable in detection of nodal and distant metastasis with accuracy comparable to PET-CT.

Ovarian cancer: PET-MRI may serve as an alternative to CT in assessing distant metastatic lesions [23,42].

### 4.7. Artificial Intelligence (AI), Deep Learning, Radiomics and Radiogenomics

AI is a relatively new science that first appeared in the early fifties. It aims to create intelligent machines with functions and reactions similar to human beings (i.e., develop theories, identification, reasoning and interpretation) with acuteness and influence typically pertaining to human [43,44]

Application of AI in the medial research is an interesting topic in modern science. In oncology, AI has been increasingly used for the diagnostic evaluation of medical images such as radiographic images, omics analysis using genome data, and clinical information [45]. AI can play an important role in different fields of gynecologic malignancies for example in medical image recognition, auxiliary diagnosis, scheming treatment and drug research.

Deep learning is a form of machine learning with more than 90% supervised learning [46]. Deep learning presents an analytical method which utilizes neural networks that employ mathematical models to imitate neuronal cells of the human brain.

Aramendía-Vidaurreta et al. described a study using a new method for automatic discrimination of ultrasound images of adnexal masses based on a neural networks approach. That method calculates seven different characteristics from the ultrasound images of the adnexal masses namely (fractal dimension, local binary pattern, invariant moments, entropy, gray level co-occurrence matrix, law texture energy and Gabor wavelet) from which several features are extracted and collected together with the patient’s age. Ultrasound images of 145 patients with adnexal masses consisting of 39 malignant and 106 benign lesions (corresponding to probability of occurrence in general population) were used to validate the purposed method in discrimination between benign and malignant adnexal masses with accuracy of 98.78%, sensitivity of 98.50%, specificity of 98.90% and area under the curve of 0.997 [47].

Many studies regarding using AI in the evaluation of medical images (e.g., MRI, hysteroscopy, colposcopy) in cervical and endometrial cancers are reported. For example, a retrospective study done by Pergialiotis et al. [48] to investigate the diagnostic accuracy of three different methodologies (Artificial neural networks ANNs, classification and regression trees CARTs and logistic regression) to predict endometrial cancer in postmenopausal women with vaginal bleeding or endometrial thickness ≥ 5 mm by ultrasound. Among the three methods, ANN analysis had the highest sensitivity, specificity and overall accuracy. The study concluded that AI is a potential powerful tool which might be used as a non-invasive screening tool to guide healthcare professionals when suspecting endometrial pathology. Another study by Urushibara et al. [49] concluded that deep learning has similar diagnostic performance to that of experienced radiologists when diagnosing cervical cancer on a single T2-weighted image. In 2019, Shen et al. [50] carried out the first study to use deep learning model for assessing 18F-FDG PET-CT images in predicting treatment outcomes in cervical cancer patients with promising results.

Thus AI is believed to improve the diagnostic efficiency, effect of treatment, overall prognosis and reduces the burden of healthcare professionals in gynecologic oncology [44].

Some difficulties for the application of AI in gynecological malignancies exit. For example, technical difficulties regarding generalizability and reproducibility, dependence on human engagement, protection of the privacy of patients and legal issues in cases of medical negligence [44,45].

Radiomics is a neoteric approach which aspires to enhance decision support by extracting quantitative information from radiologic imaging. Radiogenomics is the extension of radiomics through the combination of genetic and radiomic data [23].

Unlike radiomics, standard imaging evaluation using CT or MRI does not take into account the tumoral heterogeneity. Radiomics can be used to evaluate the whole tumor burden heterogeneity compared to single biopsy sample by extracting plenty of quantitative data which cannot be assessed visually from conventional images and integrates them with the patient’s clinical data into accessible database. The process of radiomics analysis includes image acquisition (X-ray, US, CT, MRI and PET), segmentation, feature extraction, feature selection, validation and model construction. A number of reliable algorithms currently exist that are able to segment the lesion automatically or semi-automatically. Features can then be extracted that include semantic (morphological) features and agnostic features (complex mathematically extracted quantitative features) via dedicated software. Selected extracted features are then used to generate a report that can be put in a database together with other relevant data, such as clinical and genomic data of the patient [51]. Its further details are beyond the scope of this article.

Radiomics and radiogenomics may offer a noninvasive tool for evaluation of the whole volume of the tumor, aiming at improving the prediction of patient’s outcome, optimal triage and offering best treatment options. Moreover, radiogenomics application for prognostic profiling of endometrial cancer has shown promising results [52].

Radiogenomics warrants the correlation of molecular markers (genomic data from tissue analysis and other clinical data) of disease to its imaging features thus aiming at creating affordable noninvasive imaging substitute for genetic testing that can be used in supporting treatment selection, predicting and monitoring its outcome [53].

## 5. Conclusions

Gynecologic malignancies are among the most common malignancies worldwide with significant morbidity and mortality. Advances in imaging aim at early detection, accurate staging, early recurrence detection and lesion specification. The most common gynecologic malignancies are cervical, endometrial and ovarian malignancies. Initial assessment for a patient with symptoms suspicious of gynecologic malignancy is classically US. Indeterminate lesions by US warrant further imaging usually by MRI. MRI is also the gold standard for local staging of most of gynecologic malignancies. Hybrid imaging has an established role in staging, monitoring of therapy and detection of recurrence. Radiomics and radiogenomics are potential noninvasive alternatives for evaluation of the tumors.

## Figures and Tables

**Figure 1 cancers-14-05528-f001:**
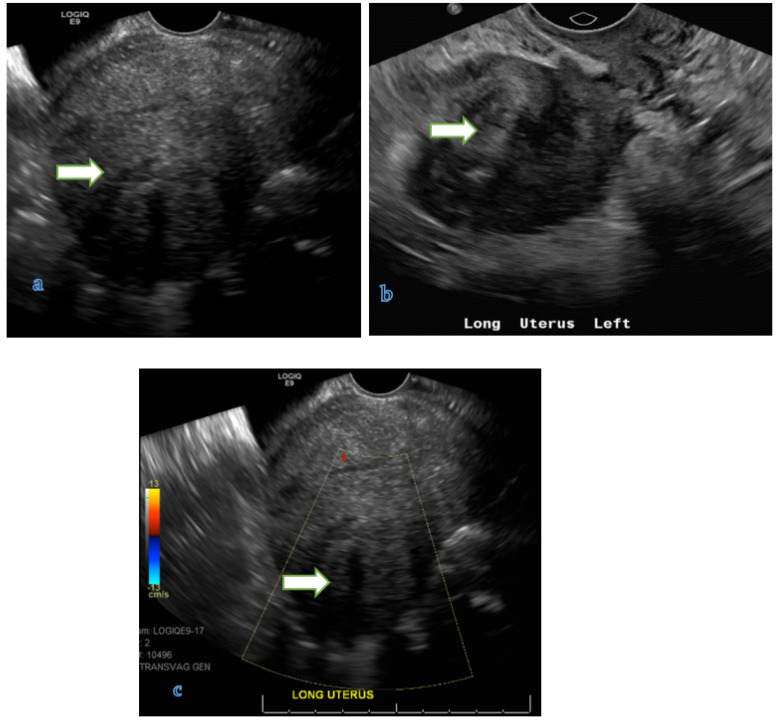
Transvaginal gray scale (**a**,**b**) and color Doppler (**c**) ultrasound images of the pelvis in a 41 year old lady with history of dysfunctional uterine bleeding show a 5.2 × 4.2 × 3.5 cm uterine mass with posterior shadowing without hypervascularity on color Doppler (B) consistent with a uterine fibroid.

**Figure 3 cancers-14-05528-f003:**
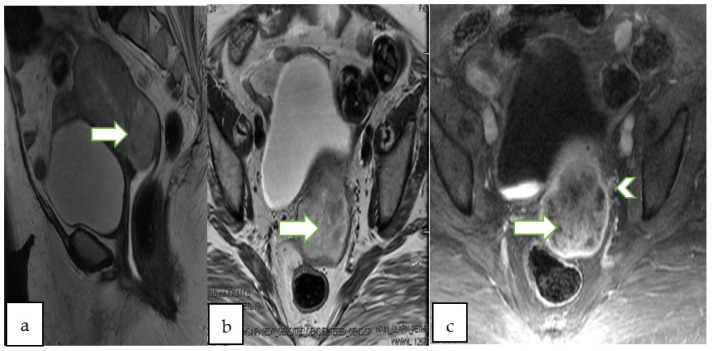
75 years old lady with post-menopausal bleeding, Sagittal T2 WI (**a**), axial T2 (**b**) and axial T1 post contrast (**c**) show a large heterogeneously enhancing infiltrative endometrial mass (arrows) extending into the cervix. Note the left parametrial extension (arrowhead). Pathology of the mass revealed endometrial adenocarcinoma (FIGO staging IIIB).

**Figure 4 cancers-14-05528-f004:**
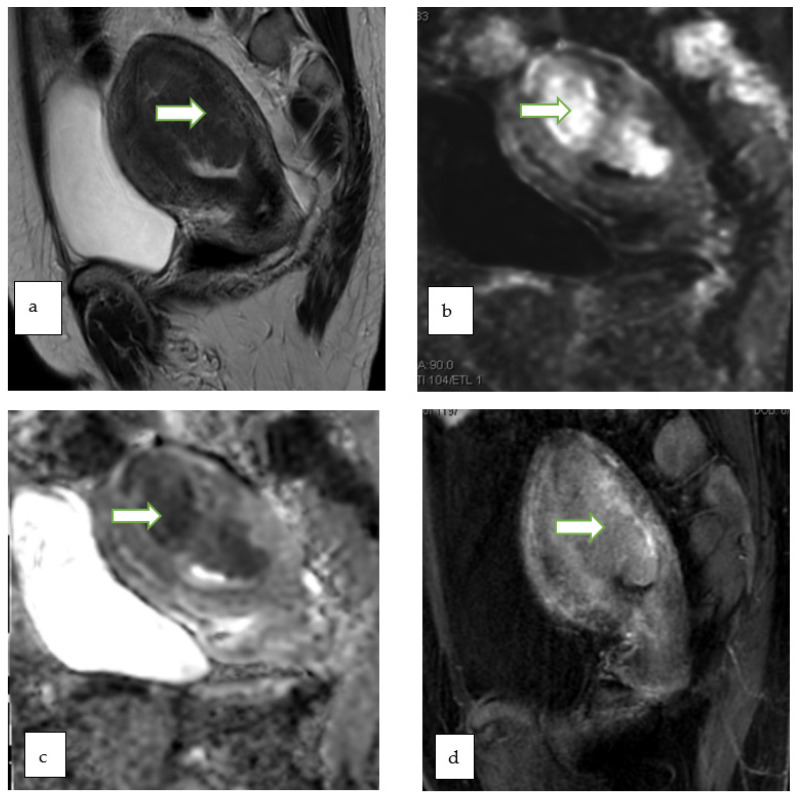
Sagittal T2W (**a**), DWI (**b**), ADC (**c**) and postcontrast fat saturated T1W (**d**) MR images of a 47 year lady with post menopausal bleeding show a hypoenhancing endometrial mass with diffusion restriction. There is <50% thick myometrial invasion along the posterior uterine wall. The patient underwent complete abdominal hysterectomy and pathology revealed endometrial endometrioid adenocarcinoma, with squamous differentiation.

**Figure 5 cancers-14-05528-f005:**
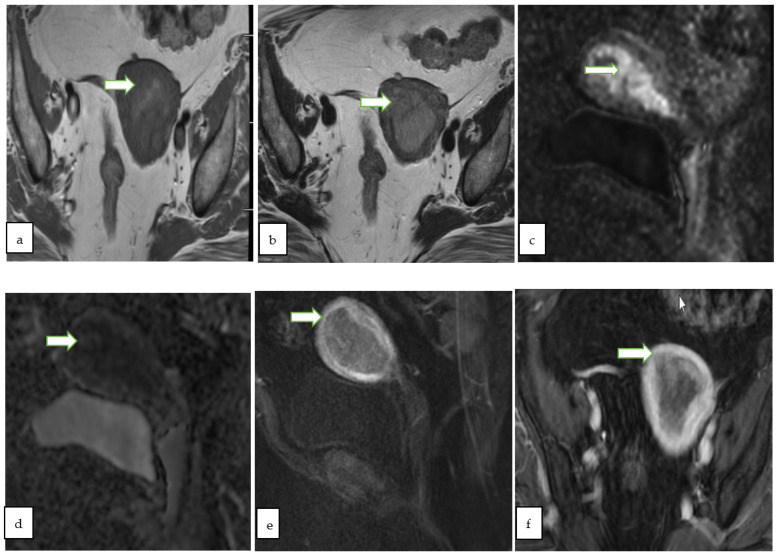
Axial T1W (**a**), axial T2W (**b**) MR images through the pelvis of a 59 year old lady with post menopausal vaginal bleeding show a predominantly hypointense expansile endometrial mass. The mass shows diffusion restriction with bright signal onsagittal DWI (**c**) with corresponding low ADC value (**d**). On sagittal (**e**) and axial (**f**) post contrast fat saturated T1W images, the mass shows >50% thick myometrial invasion of the anterior uterine wall. The patient underwent total hysterectomy and pathology revealed endometrial endometrioid adenocarcinoma.

**Figure 6 cancers-14-05528-f006:**
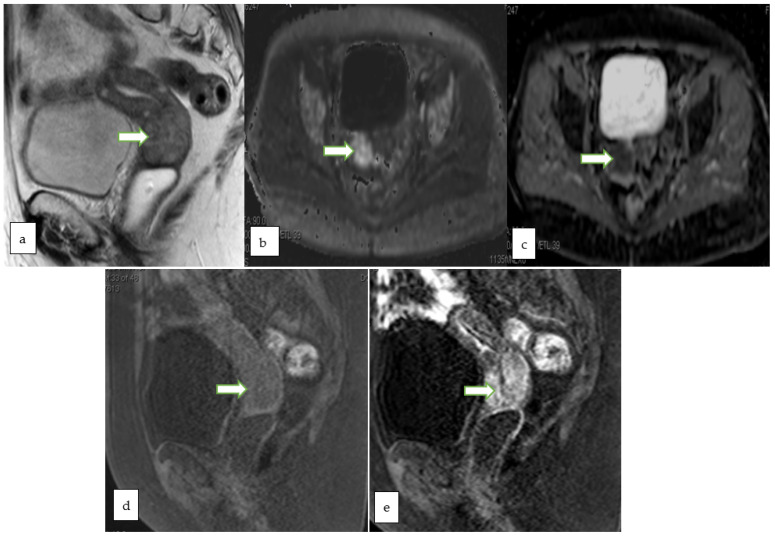
Sagittal T2W (**a**), Axial DWI (**b**) ADC (**c**) and sagittal T1 FS (**d**) MR images in a 78 year old lady with post-menopausal bleeding show lobulated expansile cervical mass involving the full thickness of the cervical stroma with intermediate signal on T2W images and restricted diffusion. Post contrast sagittal T1 FS image (**e**) shows early heterogeneous enhancement of the mass. A biopsy from the mass revealed detached fragments of squamous cell carcinoma.

**Figure 7 cancers-14-05528-f007:**
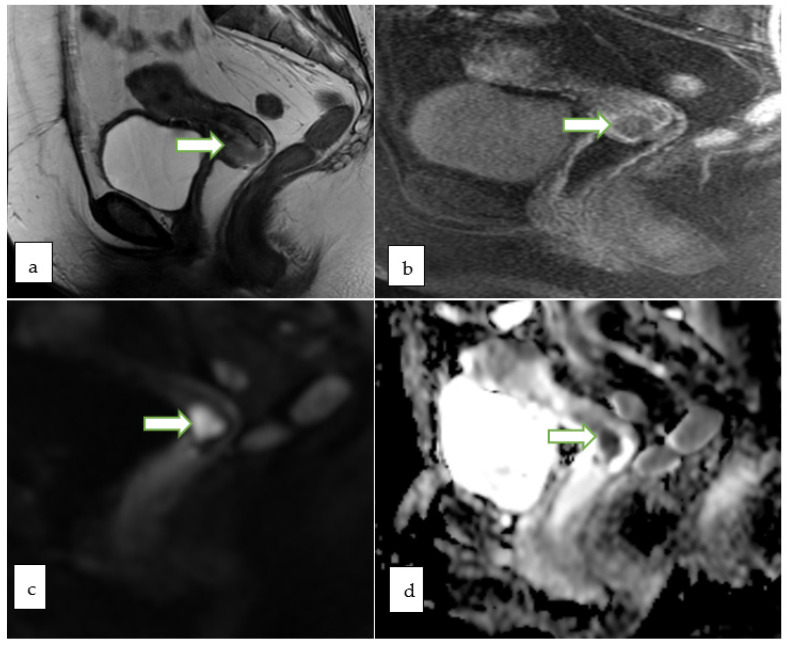
53 years old patient, MRI sagittal T2 (**a**) and sagittal T1 post contrast (**b**) showing a 2.3 × 1.6 cm endophytic mass is noted within the cervix about 1.5 cm from the internal os partially infiltrating the anterior cervical lip. The lesion appears bright in DWI (**c**) and low ADC (**d**) value denoting diffusion restriction. No evidence of parametrial, pelvic side wall or nodal involvement (Revised FIGO staging stage 1B1). Histopathology of the mass revealed moderately differentiated endocervical adenocarcinoma.

**Figure 8 cancers-14-05528-f008:**
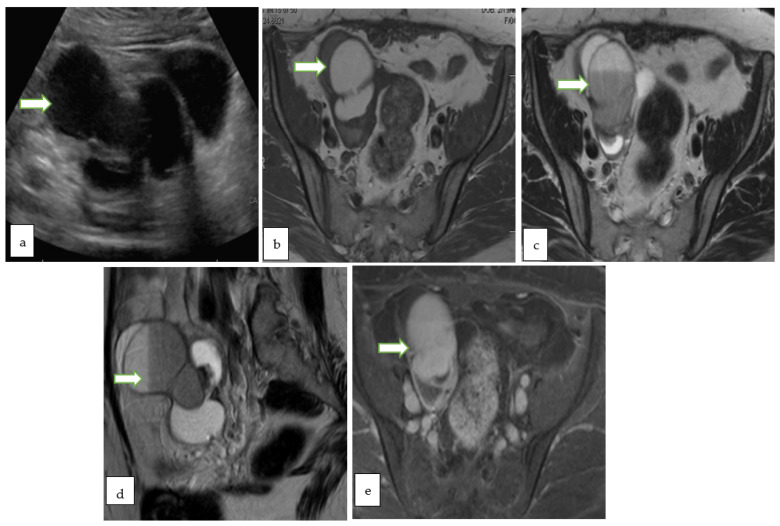
Gray scale US image (**a**) through the pelvis of a 42 year lady with non specific pelvic pain, shows a multiloculated right ovariancystic lesion with internal septations as well as low levels internal echoes (O-RADS category 3). Axial T1 W (**b**), Axial T2W (**c**), coronal T2W (**d**) and Axial T1 FS post contrast (**e**) MR images of the same case show a multiloculated cystic right adnexal lesion without enhancing soft tissue components or thick septations. The lesion demonstrates T1 hyperintense component with fluid-fluid level consistent with hemorrhagic/proteinaceous contents (O-RADS MRI 3).

**Figure 9 cancers-14-05528-f009:**
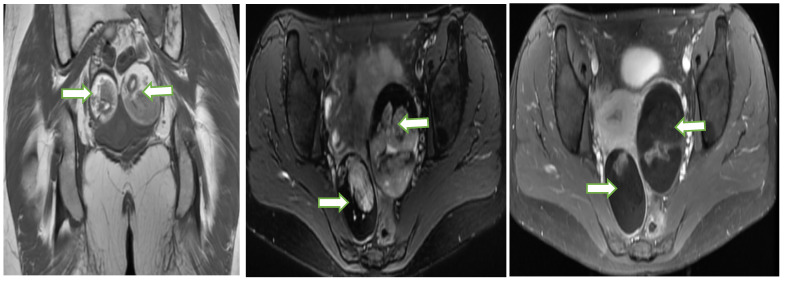
Coronal T1W (**a**), Axial T2W with fat saturation (**b**) and axial post contrast T1W (**c**) MR images through the pelvis in a 33 year old lady with abnormal vaginal bleeding show bilateral well circumscribed fat containing non-enhancing adnexal masses consistent with ovarian dermoids. Subsequent follow up ultrasound for the lesions show no significant interval changes in keeping with the benign nature of the lesions.

**Figure 10 cancers-14-05528-f010:**
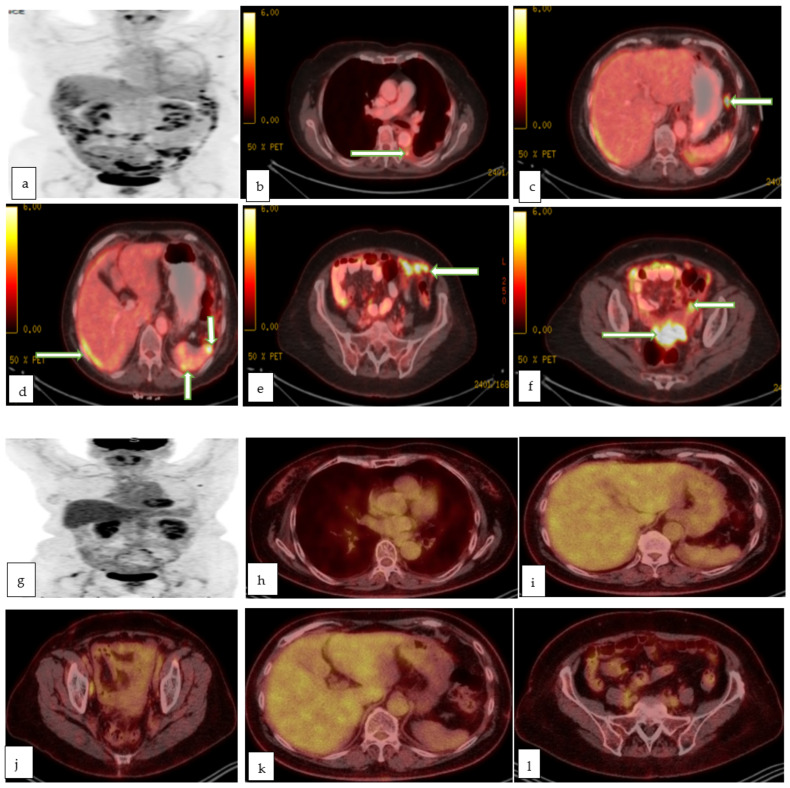
Maximum intensity projection (**a**), and multiple axial color fused images (**b**–**f**) of PET CT in a 78 year old lady with recently diagnosed left ovarian serous adenocarcinoma for staging, show left pleural effusion with hypermetabolic nodule/implant (proven to be malignant effusion) with hypermetabolic implants in the subdiaphragmatic (**c**), perihepatic and perisplenic regions (**d**), infracolic omentum (**e**) and recto-vaginal pouch (**f**) in keeping with metastasis. Follow up maximum intensity projection (**g**), and multiple axial color fused images (**h**–**l**) of PET CT in the same patient after 3 months of chemotherapy, shows excellent post-treatment response with resolution of the peritoneal metastases as well as the prior left pleural effusion.

**Figure 11 cancers-14-05528-f011:**
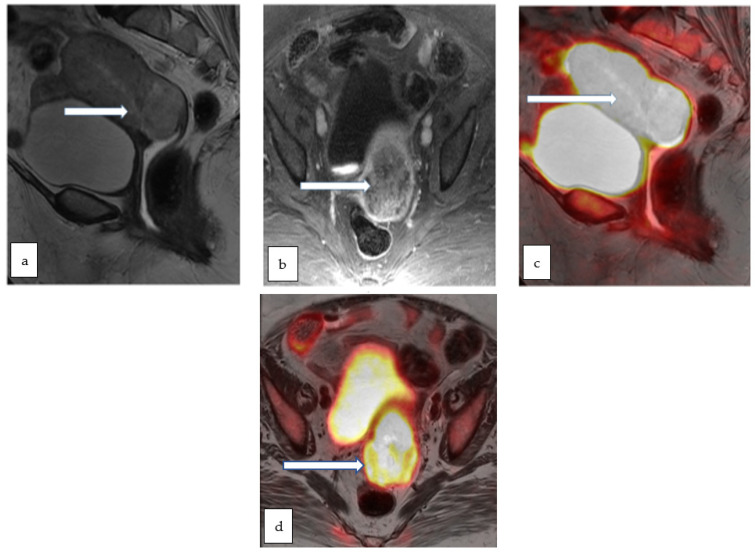
Sagittal T2 WI (**a**) and axial T1 post contrast (**b**) MR images in a 75 year old lady with post-menopausal vaginal bleeding and fatigue show a large heterogeneously enhancing infiltrative endometrial mass extending into the cervix. Corresponding sagittal (**c**) and axial (**d**) color fused PET/MR images show intensely hypermetabolic activity throughout the uterus including the cervix. Biopsy of the mass revealed high grade adenocarcinoma.

**Figure 12 cancers-14-05528-f012:**
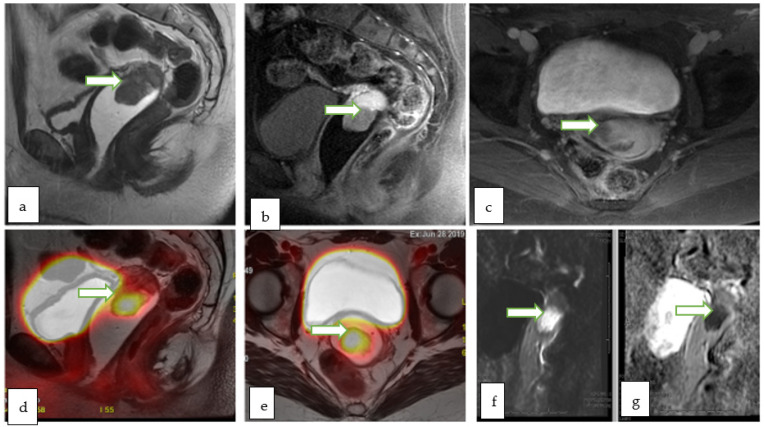
Sagittal T2W (**a**), sagittal post contrast fat saturated T1W (**b**) and axial post contrast fat saturated T1W (**c**) images through the pelvis of a 31 year old lady with incidentally discovered cervical mass during infertility workup, show an exophytic cervical mass arising from the anterior lip. The lesion appears bright in DWI (**f**) with low ADC (**g**) values consistent with diffusion restriction. Corresponding sagittal (**d**) and axial (**e**) color fused PET MRI images showed the FDG avid hypermetabolic cervical mass with maximum SUV of 13.4. There was no evidence of vaginal cuff involvement, parametrial extension or pelvic/inguinal lymphadenopathy. Biopsy of the cervical mass revealed invasive high-grade carcinoma with an adenocarcinoma component.

**Figure 13 cancers-14-05528-f013:**
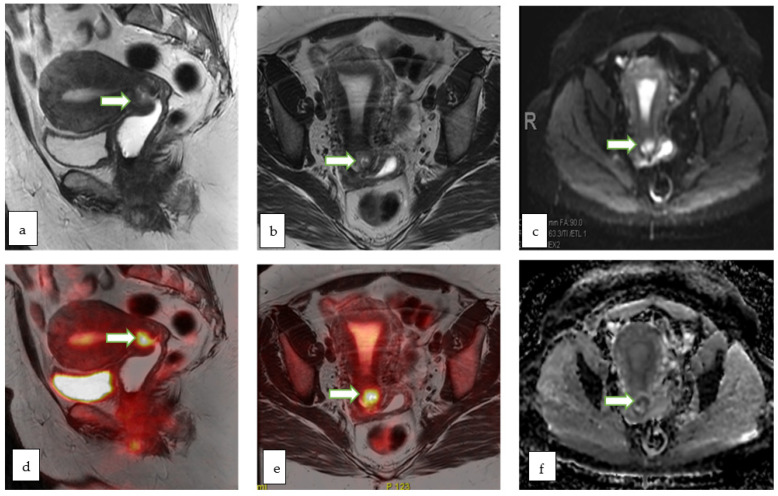
Sagittal T2W (**a**) and axial T2 W (**b**) MR images of a 51 year old lady with abnormal vaginal bleeding show a 2.3 × 2.1 cm endocervical mass involving the anterior cervical lip. The lesion appears bright on DWI (**c**) and shows low ADC (**f**) values consistent with diffusion restriction. Sagittal (**d**) and axial (**e**) color fused PET-MR images showed the corresponding FDG avid hypermetabolic with maximum SUV of 10.7. Mild uptake within the endometrial cavity was thought to be physiologic. Biopsy of the cervical mass revealed basaloid type invasive moderately differentiated squamous cell carcinoma of the cervix.

**Table 2 cancers-14-05528-t002:** O-RADS MRI Risk Stratification and Management System.

O-RADS MRI Score	Risk Category	Positive Predictive Value for Malignancy	Lexicon Description
0	Incomplete evaluation	N/A	N/A
1	Normal ovaries	N/A	No ovarian lesion
			Follicle defined as simple cyst ≤ 3 cm in a premenopausal woman
			Hemorrhagic cyst ≤ 3 cm in a premenopausal woman
			Corpus luteum +/− hemorrhage ≤ 3 cm in a premenopausal woman
2	Almost certainly benign	<0.5%	Cyst: Unilocular- any type of fluid content - No wall enhancement - No enhancing solid tissue *
			Cyst: Unilocular–simple or endometriotic fluid content -Smooth enhancing wall -No enhancing solid tissue
			Lesion with lipid content ** - No enhancing solid tissue
			Lesion with solid tissue showing dark signal on T2/DWI -Homogeneously hypointense on T2 and DWI
			Dilated fallopian tube-simple fluid content - Thin, smooth wall/endosalpingeal folds with enhancement - No enhancing solid tissue
			Para-ovarian cyst–any type of fluid - Thin, smooth wall +/− enhancement - No enhancing solid tissue
3	Low risk	~5%	Cyst: Unilocular–proteinaceous, hemorrhagic or mucinous fluid content - Smooth enhancing wall - No enhancing solid tissue
			Cyst: Multilocular-Any type of fluid, no lipid content - Smooth septae and wall with enhancement-No enhancing solid tissue
			Lesion with solid tissue (excluding T2 dark/DWI dark) - Low risk time intensity curve on DCE MRI
			Dilated fallopian tube - Non-simple fluid: Thin wall/folds - Simple fluid: Thick, smooth wall/folds - No enhancing solid tissue
4	Intermediate risk	~50%	Lesion with solid tissue (excluding T2 dark/DWI dark) - Intermediate risk time intensity curve on DCE MRI - If DCE MRI is not feasible, score 4 is any lesion with solid tissue (excluding T2 dark/DWI dark) that is enhancing ≤ myometrium at 30–40 s on non-DCE MRI
			Lesion with lipid content - Large volume enhancing solid tissue
5	High risk	~90%	Lesion with solid tissue (excluding T2 dark/DWI dark) - High risk time intensity curve on DCE MRI - If DCE MRI is not feasible, score 5 is any lesion with solid tissue (excluding T2 dark/DWI dark) that is enhancing > myometrium at 30–40 s on non-DCE MRI
			Peritoneal, mesenteric or omental nodularity or irregular thickening with or without ascites

* Solid tissue: a lesion component that enhance and conforms to one of the following morphologies: mural nodule, papillary projection, irregular wall/septation or other larger solid portions. ** Minimal enhancement of Rokitansky nodules in lesion containing lipid does not change to O-RADS MRI 4 DCE = dynamic contrast enhancement with a time resolution of 15 s or less, DWI = diffusion weighted images, MRI = magnetic resonance imaging, N/A = not applicable.

## Data Availability

No new data were created or analyzed in this study. Data sharing is not applicable to this article.

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
