# Peer review of "Recent Imaging Updates and Advances in Gynecologic Malignancies"

_cancers, 2022, doi:10.3390/cancers14225528_

Round 1

Reviewer 1 Report

In this review, the Authors aimed to provide a comprehensive evaluation of the recent imaging updates and advances in gynecologic malignancies, mainly focusing on cervical, endometrial, and ovarian cancers.

Although the manuscript is well-written, the Authors did not provide a well-organized overview of this topic for two main reasons. Firstly, it is not described in the “introduction” paragraph how the information is reported in the review. I suggest that a narrative review on such a huge topic should have a "methods" paragraph where the Authors explain how the information is displayed to the reader. Particularly, the Authors organized the review by discussing the different imaging techniques sequentially (US, MRI, CT…). However, it is not easy to understand for the reader the distinction between the different gynecologic tumors’ imaging information in each paragraph. I would recommend using more sub-paragraphs for each technique. Secondly, it is essential to have a “discussion” paragraph in which the Authors speculate about the pros and cons of each imaging technique and highlight the main concepts about how these advances may change the clinical activity of healthcare professionals involved in caring for cancer patients. In addition, it would be important that the Authors provide their perspectives on the future advances in this topic. Finally, the tables are not understandable. I suggest using the “landscape” orientation for pages with tables.

Here, I report my minor suggestions for the manuscript:

-References from the abstract should be removed. Therefore, the first reference of the paper should be the first reference in the "Introduction" paragraph

-It is essential to specify which Author is the corresponding Author

-Line 34: I suggest reporting more recent data than those of 2020

-Line 44 - 45: Remove colons

Reviewer 2 Report

MAJOR STRENGTHS

1. The text is relatively well-written, with logical formatting, and it is of appropriate length.

2. This article provides a comprehensive review of the recent advances in imaging of gynecologic malignancies with emphasis on cervical, endometrial, and ovarian neoplasms.

MAJOR WEAKNESSES

3. This article was seemed to be texture book. Thus, the readers may not be interesting.

4. It would be better to add table for describing characteristics of cervical, endometrial, and ovarian neoplasms according to imaging modalities.

5. It would be also better to add about artificial intelligence or deep learning as well as radiomics.

SPECIFIC COMMENTS

6. ultrasound vs. US vs. ultrasonography à You should use one term consistently.

7. TVUS à You should mention it initially if you used abbreviation term.

Reviewer 3 Report

Well written review

Minor corrections have been highlighted and changes elaborated.

Please also look for strike throughs at places in the attached manuscript

Reviewer 4 Report

the review “Recent Imaging updates and advances in gynecologic malignancies " furnished a useful overview of the recent advances in imaging of gynecologic malignancies.  

I'd suggest to the authors the right formatting of T1 and T2

Round 2

Reviewer 1 Report

The authors have adequately addressed my concerns. However, I suggest moving the "Discussion" paragraph after the "Radiomics" paragraph and before the "Conclusion" paragraph.